# Leaving No Women Behind: Evaluating the Impact of the COVID-19 Pandemic on Livelihood Outcomes in Kenya and Ethiopia

**DOI:** 10.3390/ijerph20065048

**Published:** 2023-03-13

**Authors:** Marshall Makate, Clifton Makate

**Affiliations:** 1Curtin School of Population Health, Curtin University, Kent Street, Perth 6102, Australia; 2School of Economics and Business, Norwegian University of Life Sciences, P.O. Box 5003, 1432 Ås, Norway

**Keywords:** SARS-CoV-2 pandemic, livelihoods, food insecurity, income, consumption, pre-existing socioeconomic inequality, female-headed household, Kenya, Ethiopia

## Abstract

The SARS-CoV-2 pandemic has revolutionised our lives, bringing with it the twin crises of illness and the need for an optimal mix of policies to alleviate its impact on the population. There needs to be more evidence on the effects of the pandemic on livelihood outcomes, including an understanding of whether female-headed families in low-income countries fare worse than their male-headed counterparts during pandemics. Using high-frequency phone surveys conducted in Ethiopia and Kenya, we examine the aggregate impact of the pandemic on income and consumption losses, as well as food insecurity. The empirical analysis estimates linear probability models that relate livelihood outcomes with household headship and other socioeconomic characteristics as controls. Overall, the pandemic increased the likelihood of food insecurity while decreasing income and consumption, particularly among female-headed households. In Kenya, living in a female-headed home increased the possibility of an adult going without food by about 10%, an adult skipping a meal by about 9.9%, and a child missing a meal by about 17% in the seven days preceding the telephone survey. In Ethiopia, living in a female-headed household increased the likelihood of an adult going hungry, skipping a meal, and running out of food by about 24.35%, 18.9%, and 26.7%, respectively. Salient pre-existing socioeconomic inequalities further exacerbated the effects of the pandemic on livelihoods. The findings have important implications for public policy and preparations by governments and other organisations interested in developing suitable gender-sensitive measures to lessen the impact of future pandemics in low- and middle-income countries.

## 1. Introduction

The ‘Global Strategy for Women’s, Children’s and Adolescents’ Health 2016–2030′ is a global strategy aimed at eradicating all preventable maternal, newborn, and child-related deaths, including stillbirths, by 2030, as well as improving women’s and children’s overall health and well-being [1]. However, as the world grapples with the devastation caused by the SARS-CoV-2 pandemic, this global commitment to ensuring women’s health and well-being is jeopardized. Women may be more vulnerable to the pandemic’s ensuing implications in several low-income countries, where social safety nets may be limited. Furthermore, we know little about the relationship between pre-existing socioeconomic circumstances and vulnerability to the pandemic and whether female-headed families fare worse than their male-headed counterparts. We investigate these issues in the context of Ethiopia and Kenya using longitudinal data collected by the World Bank to track the impact of the pandemic on livelihoods. Female-headed families are likely to fare poorly during pandemics as women may have to balance their time between childcare and employment.

To contain the virus and prevent further harm to the populace, governments around the world implemented various measures, including lockdowns, curfews, social seclusion, and wearing masks, among others. Even though such policies are implemented with the overall good in mind, they inevitably have unfavourable effects, especially when poor social safety net programmes are present. The emergence of the pandemic disrupted global food markets and, consequently, supply chains. Markets are essential in ensuring the availability and accessibility of food, which is one of the pillars of food security. In Ethiopia, value chain agents noted declining demand, clientele, and turnover, higher losses, less competition, higher transportation expenses, and modifications to the procurement landscape [2]. The pandemic had a negative impact on farmers in Kenya as well, disrupting supply chains, increasing the inflation of commodity prices, and reducing trade volumes [3,4]. Rising commodity prices limit the amount of food that can be purchased because they lower people’s purchasing power or indirectly erode their disposable income.

Recent research has emerged showing that restrictions imposed by governments to minimize the spread of the virus, such as lockdowns, are more likely to impact women in many respects disproportionately [5]. For example, women, particularly those living in low-income countries, are more likely to be impacted economically since a larger fraction is in precarious or unstable employment situations, including the informal sector, entertainment industry, arts sector, and domestic services [6]. A great number of other women are also employed in poorly paid front-line positions, for example, community health workers who continue to tackle risky tasks such as COVID-19 surveillance, contact tracing, vaccinations, and monitoring quarantine and isolation centres, and yet these essential workers are often lowly or irregularly paid with inadequate protective equipment [7,8,9]. Given the erratic nature of their income sources, people in precarious employment are more likely to experience food insecurity.

In this paper, we examine the aggregate impact of the pandemic on livelihood outcomes (income, consumption, and food insecurity) in Kenya and Ethiopia during the SARS-CoV-2 pandemic. Our empirical analysis relies on longitudinal data collected by the World Bank and its partners through high-frequency phone surveys. We contribute to the current discussions on the effect of the SARS-CoV-2 pandemic in low-income countries. First, we use high quality longitudinal data to provide insights on the gendered impact of the SARS-CoV-2 pandemic on livelihoods in Kenya and Ethiopia. Second, we show that pre-existing socioeconomic inequalities exacerbated the impact of the SARS-CoV-2 pandemic on livelihoods in Kenya and Ethiopia. These results underscore the need for governments to develop suitable gender-sensitive measures to mitigate the impact of future pandemics in Kenya, Ethiopia, and other economies with comparable structures, thus fostering an equitable and more inclusive response to future pandemics.

## 2. SARS-CoV-2 Pandemic, Gender, and Outcomes

The SARS-CoV-2 pandemic is more likely to exert social, psychological, health, and economic repercussions on communities, making some people more vulnerable to its negative effects on their livelihoods and well-being outcomes than others. Insecure housing, restricted access to health care, poverty, gender disparities, racial segregation, food insecurity, changing patterns of consumption, and loss of income and employment are among the factors influencing susceptibility to the SARS-CoV-2 pandemic and the impact of health and well-being outcomes [10,11,12]. The social determinants of health are a concept that encompasses all these aspects. The social determinants of health are defined by the World Health Organisation (WHO) as the “conditions in which people are born, grow, live, work and age” and “the fundamental drivers of these conditions” [11]. These life circumstances are, in turn, influenced by the distribution of income, power, and resources at both the local, national, and global levels. Premature death and disease are greatly influenced by social determinants of health, particularly among vulnerable groups such as women, children, the elderly, and minorities. Moreover, the impact of COVID-19 is less likely to be uniform across countries and even within the same country. Women, children, and the elderly are amongst the most vulnerable groups of the population.

Recent evidence has also shown that women living in female-headed families are at an elevated risk of experiencing poor food security and well-being outcomes [13]. Using data from India, Bau, Khanna [13] show that women from vulnerable positions within the household, including those with children, are more likely to experience poor food security and mental well-being outcomes. One of the global measures recommended to minimize the spread of the SARS-CoV-2 virus was the implementation of lockdowns. Research has emerged showing that such measures have disproportionately impacted women and girls in several aspects [5]. Women, particularly those living in low-income countries, are more likely to be impacted economically since a larger fraction are in precarious or unstable employment, which includes the informal sector, entertainment industry, arts sector, and domestic services, among others [6]. A great number of other women are employed in poorly paid front-line positions, for example, community health workers who continue to tackle risky tasks such as COVID-19 surveillance, contact tracing, vaccinations and monitoring quarantine and isolation centres, and yet these important workers are often lowly or irregularly paid with inadequate protective equipment [7,8,9].

## 3. Materials and Methods

### 3.1. Data

This study uses longitudinal data from high-frequency phone surveys conducted in Kenya and Ethiopia by the World Bank in partnership with the local governments. In Kenya, the survey is called the “rapid response phone survey (RRPS)” [14]. Survey data for Ethiopia is available up to 12 rounds, with the 12th round ending in June 2021 [15]. For our analysis, we used the first six rounds of the Ethiopian household high-frequency phone survey (HFPS). The Ethiopian household HFPS began towards the end of April 2020, and households were then called back every three to four weeks. The survey was carried out using Computer-Assisted Telephone Interviewing (CATI) in a modular fashion. The sample of Ethiopian HFPS households was drawn from the sample of Ethiopian Socioeconomic Survey (ESS) households interviewed in the 2018/2019 round. The extensive information gathered in the ESS a few months before the pandemic provides rich information and context for investigating the pandemic’s impact on livelihoods. The ESS is based on a nationally and regionally representative sample of Ethiopian households, with 6770 households interviewed in urban and rural areas. Households were asked in the survey to provide phone numbers, either their own or those of a reference household, including friends or neighbours, so that they could be contacted in the follow-up ESS surveys if they moved away from the sampled location. The sampling frame for the HFPS was made up of 5374 households that had at least one valid phone. The target sample size for the HFPS was 3300 households to obtain a representative stratum at the national, urban, and rural levels (1300 in rural and 2000 in urban areas). A detailed description of the sampling methodology and questionnaires used to collect all the data is available here [15].

The World Bank also conducted the RRPS survey in Kenya in collaboration with the Kenya Bureau of Statistics and the University of California, Berkeley. The dataset we accessed contains information from eight waves of the COVID-19 RRPS, which is part of a panel survey that began in May 2020, and the eighth wave was completed in July 2022. Even though the dataset we accessed contains eight waves of the COVID-19 RRPS, we use the first six waves for both countries to harmonise outcome variables and other key variables for the analysis. CATI techniques were used to conduct the Kenyan RRPS, just like in the Ethiopian survey. The Kenyan dataset consists of two randomly chosen samples of households that participated in the computer-assisted personal interviewing (CAPI) trial for the 2015–2016 Kenya Integrated Household Budget Survey (KIHBS) and provided a phone number [16]. The second sample of households was gathered using Random Digit Dialling technique, in which active phone numbers generated by the Kenya Communications Authority’s 2020 Numbering Frame were chosen at random. The sample, which includes both urban and rural areas, was created to be an accurate representation of Kenya’s cell phone-using population. A detailed sampling procedure including survey questionnaires is available here [14]. Kenya’s final sampling frame included 4075 households with active phones.

The purpose of these telephone surveys was to interview a nationally representative sample of households excluding those in prisons, hospitals, military barracks, and school dormitories to gauge the socioeconomic impact of the SARS-CoV-2 pandemic on livelihoods and thereby inform a targeted response. The high-frequency phone surveys collect an array of information including household background, access to basic services, employment, food security, income loss, transfers, agriculture, health, education and childcaring, and COVID-19-related knowledge among several other topics. Both surveys include a set of weights that we incorporated in our analysis to obtain unbiased estimates.

### 3.2. Measures

#### 3.2.1. Food Insecurity during the SARS-CoV-2 Pandemic

The high-frequency phone surveys asked several questions concerning the food security situation of individual households during the pandemic. The reference period for these questions was 30 days. However, questions were also asked on the number of adults and children that had either gone to bed hungry, skipped meals, or gone for entire days without food in the household and over the past seven days. Specifically, the questions asked were of the following form: “In the past seven days, how many days have adults in your household… (1) gone to bed hungry? (2) skipped meals or cut the amount of meals? (3) gone entire days without food?”. We focus on the questions that aimed to know whether the individual respondent or any other adult member from their household: (i) worried about the possibility that the household would not have enough food to eat because of a lack of money or other resources, (ii) had to go to bed hungry because of lack of money or other resources, (iii) had to skip a meal or reduce the amount of meals due to lack of money or other resources, and (iv) had run out of food because of lack of money or other resources. In Kenya, the surveys also asked whether any children in the household had skipped a meal or reduced the number of meals due to lack of money or other resources. We use the responses to these questions to create individual dummy variables that equals one if a respondent had answered yes to a question (or had indicated having one or more days they had gone hungry or skipped a meal, or they had gone entire days with no food) and zero otherwise, thereby creating outcomes reflecting hunger, food running out, adults skipping meals, children skipping meals, and worrying over the prospect of inadequate food in the household. These indicators are also consistent with those commonly used in the literature [17].

#### 3.2.2. Income and Consumption Losses during the SARS-CoV-2 Pandemic

The high-frequency phone surveys include a separate section on income loss. In Kenya, each respondent was asked whether, during the past 14 days, his/her household had to sell livestock or other household assets to generate income (such as vehicles, furniture, kitchen or electronic equipment, and tools), took a loan for use on household consumption, as well as the kind of loan they had taken (i.e., whether from a friend/relative or commercial bank, among others), had their business closed, had to rely on credit purchases, or reduced food consumption in a bid to cope with the effects of the pandemic. These questions indicate an immediate response to the impact of the SARS-CoV-2 pandemic in Kenya. In Ethiopia, the questions in round one asked respondents whether income from several sources including overall household income, business, farm, other sources, and remittances had increased, stayed the same, reduced, or was completely lost 100% following the COVID-19 outbreak. Questions in subsequent surveys (round 2 onwards), ask respondents whether, since the last phone call, income from the same sources noted earlier had increased, stayed the same, reduced, or was completely lost 100%. These questions capture the aggregate impact of the SARS-CoV-2 pandemic on income losses in Ethiopia. We use the answers to these questions to generate several dummy variables that represent income or/consumption losses. Specifically, we created a series of dummy variables to represent income or consumption loss, and these dummy indicator variables equalled one if the respondent indicated that their income or consumption had decreased or was completely lost because of the pandemic, and zero otherwise. For consumption losses and income losses, separate variables are generated.

### 3.3. Empirical Strategy

The study evaluates the gendered impact of the SARS-CoV-2 pandemic on livelihoods (income, consumption, and food insecurity) and inequity in access to health services and necessities in Kenya and Ethiopia. In this study, we use the gender of the head of household as an indicator for gender. We test whether individuals living in female-headed families fare worse when compared to those living in male-headed families. To fulfil the aims of the study, the empirical analysis proceeds in two steps. First, we explore the relationship between household headship and livelihood outcomes accounting for variables that reflect potential vulnerability within households in Kenya and Ethiopia. To this end, we assess whether women from female-headed households experience worse outcomes during the pandemic compared to their counterparts in male-headed families. In this instance, vulnerability relates to those individuals from families with children, low levels of education, and lower pre-COVID-19 socioeconomic status level. The model we estimate takes the following form:(1)$$Y_{iarct}=\beta_{1}children_{i}+\beta_{2}female\_head_{i}+a_{a}+\delta_{r}+w_{t}+\Gamma X_{i}+\varepsilon_{iarct}$$
where i represents the individual respondent, a represents her age, r represents her region of residence, c is her county of residence, $Y_{iarct}$ measures the outcome variable (income losses, lost consumption, and food insecurity outcomes), $children_{i}$ is a categorical variable with four categories and representing the number of children in the household (1 = no children, 2 = 1–2 children, 3 = 3–4 children, and 4 = 5 or more children), and $female\_head_{i}$ denotes whether the respondent resides in a family or household where the head is female. Previous research suggests that households with children are more likely to experience food insecurity [18,19]. In all the estimated specifications, we include age-fixed effects $a_{a}$, region-fixed effects $\delta_{r}$, and survey-fixed effects $w_{t}$. The vector of additional control variables $X_{i}$ incorporates the respondent’s level of education (no education, primary education, and secondary or higher education) since a higher level of schooling is associated with a lower prospect of food insecurity [20]. We also include controls for whether the respondent had lost a job during COVID-19, and lives in a rural or urban area. To minimize omitted variable bias, we include pre-pandemic socioeconomic status represented by an asset index with five quintiles (poorest (quintile 1), poorer, average, richer, and richest (quintile 5)). We follow Filmer and Pritchett [21] in generating a household asset index using principal components analysis (PCA). Survey respondents were asked questions regarding ownership of several assets prior to the pandemic, including radio, mattress, charcoal jiko, refrigerator, television, landline, or computer/laptop before March 2020. In further analysis, we use this pre-pandemic household wealth measure to assess whether the pandemic had a greater impact on households who were already in more vulnerable positions. The standard errors, $\varepsilon_{iarct}$, are clustered at the county or enumeration area level.

### 3.4. Identifying the Impact of the SARS-CoV-2 Pandemic

The study evaluates the impact of the SARS-CoV-2 pandemic on livelihood outcomes in Kenya and Ethiopia. We specifically test whether the pandemic had a disproportionately larger impact among female-headed families when compared to male-headed families. To identify the impact of the SARS-CoV-2 pandemic, we rely on the nature of the questions asked in the high-frequency phone surveys that allow us to measure the aggregate impact of the pandemic as these relate to the changed circumstances following the onset of the pandemic. For Kenya, since we were unable to identify or match the pre-COVID-19 data to the COVID-19 high-frequency phone survey data, identifying the impact of the pandemic is somewhat complicated for some of our outcomes particularly those relating to food insecurity. The questions asked in the surveys refer to the prevailing situation or changed circumstances of the household in the past one week or two weeks within the last 30 days. This is a much shorter reference window and suggestive of the changed circumstances of the household following the emergence of the pandemic. We interpret these findings on the impact of the pandemic on food insecurity bearing in mind that what we document are mere associations indicating the prospect of experiencing poor outcomes among individuals from female-headed families during the SARS-CoV-2 pandemic. However, in some of the food insecurity outcomes, the survey question allows us to clearly identify the aggregate impact of the pandemic. For example, respondents were asked the following question: “Compared to before March 2020, before the lockdown/pandemic, are you more/less/equally worried about your household not having enough food?”.

For outcomes linked to income losses and difficulty with access to necessities, we can clearly identify the aggregate impact of the pandemic. In Kenya, the survey questions asked respondents on the specific coping strategies that the household adopted in response to the income losses prompted by the SARS-CoV-2 pandemic (as described earlier). The coping strategies such as the selling of personal assets, borrowing from friends or relatives, taking loans from financial institutions, delaying payment obligations, credited purchases, business closures, and reducing food consumptions all clearly reflect the aggregate income losses following the onset of the pandemic. In Ethiopia, the questions on income losses clearly capture the aggregate impact of the pandemic as these reflect the changes in income from the onset of the pandemic and measured at a specific point in time.

## 4. Results

### 4.1. Summary Statistics

Table 1 reports the weighted summary statistics for selected variables and using only the first wave of each country. Using the data for the first wave of the high-frequency surveys for either county, we observed that the proportion of female-headed families was 24.38% and 30.09% in Ethiopia and Kenya, respectively. The average age of respondents was higher in Ethiopia 39.01 vs. 35.30 years in Kenya. The proportion of households with no children was 22.24% in Ethiopia compared to 31.55% in Kenya. Most of the households in both countries had at least 1–2 children, with 45.33% in Ethiopia compared to 41.33% in Kenya. We observed a smaller fraction of households reporting to have five or more children, 8.07% in Ethiopia compared to 5.22% in Kenya. In terms of socioeconomic status of households prior to the SARS-CoV-2 pandemic, 16.69% compared to 26.88% of the households were classified as the poorest (asset quintile 1) in Ethiopia and Kenya, respectively. In Kenya, only 6.47% of the households were classified as the richest (asset quintile 5) compared to 22.96% in Ethiopia. The fraction of households living in rural area was comparable at 66.89% in Ethiopia and 66.52% in Kenya.

### 4.2. SARS-CoV-2 Pandemic and the Vulnerability of Women within the Household

The results in Table 2 and Table 3, estimated using Equation (1), speak to the vulnerability of women during the SARS-CoV-2 pandemic in Kenya. In this instance, we explore the relationship between family structure and food insecurity outcomes including outcomes linked to income and consumption losses during the pandemic in Kenya. We show that women who are in more vulnerable positions within the household were more likely to fare worse compared to their counterparts from male-led families during the pandemic. When the head of household is female, the probability of going hungry during the pandemic increased by an estimated 3.96 percentage points (pp) and is statistically significant at the 1% level. Given that the mean of the outcome variable in our analysis sample was 39.40%, the 3.96 pp effect represents an approximate 10% $\left(\frac{0.0396}{0.394}\right)\times 100$ increase in the probability that the respondent or an adult from a female-headed family would go to bed hungry in the past seven days and during the pandemic in Kenya. Households with five or more children were 14.33 pp more likely to have any adult go hungry when compared to households with no children and statistically significant at the 1% level. There is a clear positive gradient suggesting that the prospect of hunger increases with the number of children in the household. The probability of an adult going hungry is exacerbated if an adult member from the household had been laid off or lost their job involuntarily since January of 2020. Having lost a job due to COVID-19 was associated with a 10.92 pp increase in the probability of going hungry during the pandemic period. Additionally, the pre-COVID-19 socioeconomic status of the household, as measured by the household wealth index, is an important determinant of the probability of going hungry. We observed that, compared to families that were classified as richest (asset quintile 5), families in the bottom poorest quintiles are more likely to experience hunger during the pandemic. Being in the poorest asset wealth group (quintile 1) before the pandemic was associated with a 16.33 pp increase in the probability of going hungry during the pandemic. The effects on hunger are also compounded when family resided in a rural area as compared to an urban locality (4.18 pp). 

Table 2 also indicates that when the respondent lives in a female-headed family, she or another adult from the same household is 3.42 pp more likely to skip a meal or cut the number of meals eaten in the past seven days prior to the survey and during the pandemic. The 3.42 pp effect represents an imprecise 9.88% increase in the probability that an adult from a female-headed family skips a meal. The effects on the prospect of skipping meals or cutting the number of meals eaten are also exacerbated when the family has children compared to when there are no children, when a household member had lost a job during the pandemic, the family lives in a rural area, and the family is of relatively low wealth compared to other families classified as rich prior to the pandemic. When the respondent lived in a female-led family, a child is 3.33 pp more likely to skip a meal. Given that the average probability that a child skipped a meal in our Kenyan sample was 19.40%, the 3.33 pp represents an approximate 17.16% increase in the probability that a child had skipped a meal in the seven days prior to the phone survey and during the pandemic. The probability that a child skips a meal is further exacerbated when the family has children in the household, someone in the family had lost a job, the family lives in a rural area, and the family was relatively poor prior to the pandemic. The results also show that, when the respondent lives in a female-headed family, she is 3.41 pp more likely to worry about not having enough food compared to the period before March 2020 and before the lockdown or pandemic. Since 53.50% of the respondents in the analysis sample indicated that they were increasingly more worried that their household would not have enough food, the 3.41 pp effect represents an imprecise 6.37% increase in anxiety over the prospect of the family not having enough food compared to the pre-COVID-19 period.

Table 3 reports the results from estimating Equation (1) using the analysis sample for Kenya. Here, we are interested in examining whether families headed by a female respondent fare poorly in terms of income losses during the pandemic in Kenya. We include the same set of controls as in Table 2. The results show that when the respondent lives in a female-headed family, there is a 2.4 pp decline in the probability that the household would have sold livestock to cope with the effects of the pandemic. This result is statistically significant at the 1% level.

The point estimates in Table 3 also indicate that being from a female-led household was associated with a 1.20 pp increase in the probability of taking a loan for use on household consumption in the past 14 days prior to the phone survey. Given that the mean of the dependent variable was 8.3% in our sample, the 1.20 pp effect represents an imprecise 14.46% increase in the probability of taking a loan for use on household consumption. When the respondent lived in female-led family, the prospect of borrowing from friends increased by 2.49 pp and is statistically significant at the 1% level. Female-headed families were 0.46 pp less likely to sell other assets during the pandemic in Kenya. The point estimates also show that female-led families were 6.51 pp more likely to report having closed a business due to effects of lockdowns or curfews and 3.88 pp more likely to rely on credit purchases to cope with the effects of the pandemic. A female-led family was 3.03 pp more likely to report reducing food consumption during the pandemic and is statistically significant at the 1% level. Given that the mean of the dependent variable was 45.6% in our sample, the 3.03 pp effect represents an approximate 6.64% reduction in food consumption among female-headed families in Kenya during the SARS-CoV-2 pandemic.

Table 4 reports the point estimates from estimating Equation (1) using the data for Ethiopia. Living in a female-headed family is associated with a 2.07 pp increase in the probability that the respondent or an adult from their household went hungry and did not eat because there was not enough money or other resources for food in the past 30 days prior to the survey. Given that the mean of the dependent variable in our sample was 8.5%, the 2.07 pp effect represents an imprecise 24.35% increase in the prospect of going hungry during the pandemic in Ethiopia. The effects of the likelihood of hunger are also exacerbated when there are children in the household, the respondent or other household member had lost a job due to the pandemic, and the respondent was from a relatively poor family prior to the pandemic. These effects are statistically significant. Being able to read and write, reside in rural area, and being single appear to be negatively associated with the prospect of going hungry. The next column reports the point estimates exploring the likelihood that an adult would skip a meal during the pandemic. When the respondent lives in a female-headed family, she or another adult from their household is 4.42 pp more likely to have skipped a meal in the past 30 days prior to the telephone survey and during the pandemic in Ethiopia. This 4.42 pp effect represents an imprecise 18.89% increase in the probability of skipping a meal during the pandemic. The prospect of worrying that the household would not have enough food to eat because of lack of money was 7.2 pp higher among female-headed families compared to their counterparts. This 7.2 pp effect represents an imprecise 17.85% increase in the probability of anxiety over not having enough money or other food resources in the household in Ethiopia and during the pandemic. When the respondent comes from a female-headed household, the probability that the household had run out of food because of lack of money in the 30 days prior to the survey was 5.36 pp higher when compared to male-headed families. This effect represents an approximate 26.67% increase in the probability that the household would run out of food due to lack of money in the 30 days prior to the phone survey. The prospect of not eating for a day was 1.53 pp higher among female-headed households, representing an approximate 21.86% increase in the likelihood of not eating for the entire day. The observed effects were also exacerbated when the respondent came from a household with children, had lost a job due to SARS-CoV-2 crisis, and was from a relatively poor family before the pandemic started. 

In Table 5, we report the aggregate impact of the pandemic on income in Ethiopia and assess whether female-headed families fare worse compared to their male-headed counterparts. Each respondent in the Ethiopian phone survey was asked the following question: “Since [LAST CALL], has income from [SOURCE] increased, stayed the same, reduced, or total 100 loss?”. We use responses to this question to indicate loss (reduced or 100% loss) of total household income, business income, farm income, lost income from other sources, and lost remittances. The results show that being from a female-headed family was associated with a 2.73 pp increase in the probability of losing overall or total household income during the pandemic in Ethiopia. Given that the average fraction of households reporting losses in total household income was 38.8%, the 2.73 pp effect represents an approximate 7.04% decline in total household incomes during the pandemic in Ethiopia. The effect on lost business income is also exacerbated when the head of the family is female when we control for pre-pandemic socioeconomic status. When the respondent lives in a female-led household, the probability of reporting reduced business income was 1.29 pp, representing an approximate 53.75% decline in the probability of reporting reduced business incomes among female-headed families in Ethiopia and during the pandemic. Female-headed families also reported having lost remittances, a 1.41 pp increase, representing an approximate 60.04% decline in the probability of remittance income.

### 4.3. What Explains the Vulnerability of Female-Headed Families in Kenya and Ethiopia during the SARS-CoV-2 Pandemic?

This study has shown that in Kenya and Ethiopia, the pandemic disproportionately negatively affected people who lived in female-headed households. We explored several pre-pandemic socioeconomic characteristics of households in these countries that could make female-headed families more vulnerable than their male-led counterparts. We consider factors such as education level, access to health insurance, literacy, marital status, and ownership of assets such as radio, television, and kitchen appliances. We present these results in a Appendix A. In Appendix A, we noted that a female-headed family in Kenya was less likely to report owning a radio (6.60 pp), television (12.80 pp), or a computer/laptop/tablet (2.10 pp). The results also indicate that being from a female-headed family was associated with a 3.30 pp decrease in the probability of having completed secondary school but a 3.90 pp higher likelihood of having internet access at home. All the results were statistically significant. For Ethiopia, we noted that the likelihood of owning a radio, television, gas stove, electric stove, and refrigerator declined significantly. Additionally, the prospect of being divorced was much higher (44.90 pp) among individuals from female-headed families. The probability of being married, literate, ever attending school, and of having health insurance was also significantly lower among female-headed families. For example, being from a female-headed family in Ethiopia was associated with a 14.20 pp decline in the probability of being able to read and write. The findings suggest that the SARS-CoV-2 pandemic’s impact was exacerbated by the pre-existing inequality in socioeconomic circumstances of female-headed families in Kenya and Ethiopia.

## 5. Discussion

The primary objective of this paper was to investigate the impact of the SARS-CoV-2 pandemic on livelihood outcomes and ascertain whether female-headed families fare worse than their male-headed counterparts in Ethiopia and Kenya. The pandemic was associated with increased food insecurity and decreased incomes and consumption, particularly among female-headed families. Our results are consistent with emerging literature in low-income countries (see, for example, [22,23]). Given the unequal livelihood implications, the findings highlight the need for governments to adequately prepare for future pandemics by developing strategies that are gender sensitive. 

Our results indicate that individuals in more vulnerable positions within the household, those with children, and those of low socioeconomic status before the SARS-CoV-2 pandemic were likelier to experience worse food insecurity outcomes, lose incomes, and reduce food consumption during the pandemic. Indeed, previous evidence across several countries, including the United States, suggests that households with children are more likely to experience food insecurity during pandemics than those without children [24]. More children, thus more mouths to feed, increases intra-household competition for food resources [25]. Given that several governments across the globe implemented policies to contain the pandemic, such as lockdowns and curfews, several households bore undesirable consequences through food insecurity, lost incomes, and lower consumption. In Kenya and Ethiopia, where a significant fraction of women was in precarious employment, losing incomes and wage employment was inevitable and had far-reaching implications on food insecurity [26,27,28,29].

The finding that female-headed families from low socioeconomic status were more likely to experience worse food insecurity outcomes reflects the vulnerability of female-headed families in low-income countries. The observation that female-headed families were generally poorer before the pandemic is consistent with economics studies contending that female-headed families in low-income countries are, on average, poorer [30,31]. The additional analysis further supports this finding we conducted that shows that these families are less likely to own even the very basic of assets such as a radio, television, cooking stove, and refrigerator, among others. Household wealth indicates a family’s material well-being distinct from expenditure and may reflect the vulnerable households [32]. The lack of asset wealth increases a household’s vulnerability to shocks such as the SARS-CoV-2 pandemic in that they will have no assets to draw on or at least sell for their own consumption smoothing needs. The latter was confirmed in Kenya (though the result was not statistically significant). We found that female-headed families were more likely to have had no assets before the pandemic when could then explain their heightened vulnerability during the pandemic. Overall, the finding that female-headed families from low-socioeconomic positions are more likely to be vulnerable to the pandemic suggests that governments in low-income countries should prioritise targeting impoverished female-headed families living with children for a more inclusive and equitable response to future pandemics.

Our analysis also reveals the negative implications of the pandemic on consumption and income. In both countries, to cope with the adverse effects of the pandemic, most households lost consumption by skipping meals, while others responded through credit purchases. The impacts on consumption are more likely to be explained by the fact that many families in our sample reported losing employment, closing businesses, and having no remittances, contributing to reduced household income. We also show that pre-existing socioeconomic inequalities have exposed families in terms of food insecurity, income, and consumption. Additionally, families from lower socioeconomic quintiles had a higher likelihood of experiencing worse livelihood outcomes. These results underscore the need for governments to realise that a one-size fits all approach will not be an effective way to deal with future pandemics. These results also assist the government in prioritising public resources and focusing on deprived areas and vulnerable groups of the population, such as women and children, so as not to leave anyone behind and hence focus on a more inclusive response [25,33].

Female-headed families in Kenya and Ethiopia were also more likely to have reported reduced household income, closed their businesses, and lost remittances. One possibility is that female-headed households are likely to lose their primary source of income during the pandemic since most women in low-income countries, including Ethiopia and Kenya, work in precarious employment arrangements and, hence, are more likely to lose their jobs—the primary source of their income [34]. The informal sector is relatively large in Kenya, employing at least 70% of workers, with women comprising at least 66% [35]. In Ethiopia, women are also overrepresented in the informal sector [36]. The precarious nature of such employment arrangements in the absence of social protection or where social protection is weak or slow to react makes this group of women even more vulnerable. Studies in Kenya’s informal settlements demonstrated that an estimated 43% of people lost all their income as the pandemic unfolded [34]. These results are also consistent with early research in developing countries showing that the SARS-CoV-2 pandemic had an unequal impact among individuals, with women bearing the more significant brunt [37]. In several countries across the globe, women were reported to have had difficulties coping with the pandemic [31]. This finding demonstrates that pandemics such as that of SARS-CoV-2 are more likely to generate and exacerbate economic inequality. Given the lower state capacity to deal with impending shocks in many low-income countries, female-headed families and other vulnerable groups are at increased risk of experiencing food insecurity [37].

There are a few limitations to our analysis. First, our analysis does not uncover a causal relationship between the SARS-CoV-2 pandemic and livelihood outcomes. Our results could be subject to selection bias since our data is only nationally representative of the population using telephones or mobile phones in Kenya and Ethiopia. Second, many data sets, including high-frequency phone surveys, are still being collected only at the household level, and this constitutes a limitation for proper gender analysis. In this instance, we cannot know the intra-household food allocations, nor which specific family members in the household are going hungry or skipping a meal or their gender. Nevertheless, our analysis provides valuable insights into the gendered impact of the SARS-CoV-2 pandemic in Kenya and Ethiopia.

## 6. Recommendations

This study has provided new empirical evidence for Kenya and Ethiopia regarding the impact of the SARS-CoV-2 pandemic on livelihood outcomes. The results point towards the vulnerability of female-headed families during the pandemic. Notably, we show that pre-existing socioeconomic inequalities further exacerbate the impact of the pandemic, with female-headed families bearing the brunt of its effects. There are several policy options available that governments could use to better prepare for future pandemics. The ideas we suggest here are not short-term interventions, but rather medium to longer term options to address pre-existing socioeconomic vulnerabilities. Considering our results, there is a need for governments in Kenya and Ethiopia to craft policies that deliberately target the most vulnerable groups of the population, especially female-headed families living with children and families from low socioeconomic positions to promote an equitable response to future pandemics and maximise social protection. Additionally, governments with a forward-looking perspective could prioritise policies that promote growth and development across low-income communities to reduce poverty and, as a result, socioeconomic inequalities over time, thus building resilience to future pandemics. Fundamentally, policy responses to the crisis must incorporate a gender lens and consider women’s particular needs, obligations, and perspectives. 

## 7. Conclusions

In this paper, we studied the implications of the SARS-CoV-2 pandemic on food insecurity, incomes, and consumption using high-frequency telephone survey data from Ethiopia and Kenya. We show that female-headed families, including those with children and from lower socioeconomic status before the pandemic, were disproportionately impacted by the pandemic. From a policy standpoint, our findings highlight the importance of clearly understanding pre-existing socioeconomic circumstances to formulate an inclusive and equitable response to impending pandemics. Future research should investigate the impact of pre-existing socioeconomic inequalities on income-related disparities in access to healthcare services and necessities during pandemics in low- and middle-income countries.

## Figures and Tables

**Table 1 ijerph-20-05048-t001:** Weighted summary statistics for selected variables (using wave 1 data for Kenya and Ethiopia).

	Ethiopia		Kenya	
Variables	N	Mean (%)	N	Mean (%)
Age (years)	3249	39.01	10,374	35.30
Female-headed family	995	24.38	2733	30.09
Able to read and write	2440	58.18		n/a
Never married	536	13.62		n/a
Married	2120	71.98		n/a
Divorced/widowed	547	14.40		n/a
Number of children in the household				
None	1107	22.24	3014	31.55
1–2 children	1459	45.33	4346	41.33
3–4 children	533	24.36	2305	21.89
5 or more children	150	8.07	709	5.22
pre-COVID-19 household wealth				
asset quintile 1 (poorest)	220	16.69	2912	26.88
asset quintile 2	217	18.60	1746	14.54
asset quintile 3	373	19.30	5082	52.10
asset quintile 4	649	22.45		n/a
asset quintile 5	1751	22.96	634	6.47
Rural resident	966	66.89	34,759	66.52
Observations	3249		10,374	

Note: Data comes from the first wave of the high-frequency phone surveys for Kenya and Ethiopia.

**Table 2 ijerph-20-05048-t002:** Relationship between household structure and food insecurity outcomes during the pandemic in Kenya.

	Went Hungry		Adult Skipped Meals		Child Skipped Meals		Worry over Inadequate Food	
Specification	β	Std. Error	β	Std. Error	β	Std. Error	β	Std. Error
Female-headed family	0.0396 ***	0.0048	0.0342 ***	0.0047	0.0333 ***	0.0037	0.0341 ***	0.0104
Number of children								
1–2 children	0.0648 ***	0.0053	0.0719 ***	0.0052	0.1517 ***	0.0037	0.0637 ***	0.0114
3–4 children	0.1126 ***	0.0064	0.1218 ***	0.0063	0.2140 ***	0.0050	0.1009 ***	0.0132
5 or more children	0.1433 ***	0.0105	0.1443 ***	0.0104	0.2559 ***	0.0095	0.0818 ***	0.0209
Lost a job during COVID-19	0.1092 ***	0.0134	0.1459 ***	0.0133	0.0991 ***	0.0122	0.0708 ***	0.0188
pre-COVID-19 household wealth								
asset quintile 1 (poorest)	0.1633 ***	0.0098	0.1575 ***	0.0094	0.1016 ***	0.0070	0.2450 ***	0.0206
asset quintile 2	0.1863 ***	0.0101	0.1434 ***	0.0097	0.1007 ***	0.0072	0.1762 ***	0.0216
asset quintile 3	0.0994 ***	0.0098	0.0951 ***	0.0095	0.0516 ***	0.0073	0.1661 ***	0.0196
asset quintile 4	0.1325 ***	0.0102	0.1114 ***	0.0098	0.0626 ***	0.0068	0.0000	.
Rural resident	0.0418 ***	0.0046	0.0130 **	0.0045	0.0189 ***	0.0036	−0.0218 *	0.0094
Age-fixed effects	Yes		Yes		Yes		Yes	
Region-fixed effects	Yes		Yes		Yes		Yes	
Survey-fixed effects	Yes		Yes		Yes		Yes	
Mean of the dependent variable	0.394		0.346		0.194		0.535	
Number of observations	46,129		46,134		46,134		11,292	

Notes: *** Significant at 1% level; ** significant at 5% level; * significant at 10% level.

**Table 3 ijerph-20-05048-t003:** Relationship between household structure and livelihoods (lost incomes or consumption) in Kenya.

	Sold Livestock		Took a Loan		Borrowed from Friends		Business Closed		Credited Purchases		Reduced Food Consumption	
Specification	β	Std. Error	β	Std. Error	β	Std. Error	β	Std. Error	β	Std. Error	β	Std. Error
Female-headed family	−0.0240 ***	0.0025	0.0120 **	0.0044	0.0249 ***	0.0042	0.0651 *	0.0298	0.0388 ***	0.0064	0.0303 ***	0.0077
Age-fixed effects	Yes		Yes		Yes		Yes		Yes		Yes	
Region-fixed effects	Yes		Yes		Yes		Yes		Yes		Yes	
Survey-fixed effects	Yes		Yes		Yes		Yes		Yes		Yes	
Mean of the dependent variable	0.075		0.083		0.071		0.262		0.210		0.456	
Observations	46,134		21,596		21,596		1097		21,596		21,596	

Notes: *** Significant at 1% level; ** significant at 5% level; * significant at 10% level. All regressions include indicators for the number of children in the household, having lost a job during the pandemic, pre-pandemic household wealth level, and living in a rural area.

**Table 4 ijerph-20-05048-t004:** Relationship between household structure and livelihood outcomes linked to food insecurity in Ethiopia.

	Went Hungry		Adult Skipped Meals		Worry Inadequate Food		Food Ran Out	
Specification	β	SE	β	SE	β	SE	β	SE
Female-headed family	0.0207 ***	0.0062	0.0442 ***	0.0092	0.0721 ***	0.0106	0.0536 ***	0.0083
Number of children								
1–2 children	0.0219 ***	0.0059	0.0419 ***	0.0085	0.0794 ***	0.0099	0.0570 ***	0.0077
3–4 children	0.0393 ***	0.0082	0.0495 ***	0.0116	0.0705 ***	0.0132	0.0673 ***	0.0102
5 or more children	0.0784 ***	0.0155	0.0714 ***	0.0201	0.1586 ***	0.0213	0.0969 ***	0.0174
Lost a job during COVID-19	0.0861 ***	0.0226	0.1071 ***	0.0288	0.1951 ***	0.0305	0.1327 ***	0.0187
Able to read and write	−0.0444 ***	0.0075	−0.1203 ***	0.0107	−0.1405 ***	0.0113	−0.0660 ***	0.0088
pre-COVID-19 household wealth								
asset quintile 1 (poorest)	0.0777 ***	0.0123	0.2054 ***	0.0187	0.1693 ***	0.0199	0.0575 ***	0.0143
asset quintile 2	0.1072 ***	0.0133	0.1788 ***	0.0187	0.1790 ***	0.0199	0.1053 ***	0.0154
asset quintile 3	0.1243 ***	0.0109	0.2084 ***	0.0144	0.2175 ***	0.0153	0.1338 ***	0.0124
asset quintile 4	0.0882 ***	0.0077	0.1561 ***	0.0105	0.1485 ***	0.0116	0.1174 ***	0.0091
Marital status								
never married	−0.0336 ***	0.0079	−0.0585 ***	0.0113	−0.1017 ***	0.0133	−0.0678 ***	0.0100
divorced/separated/widowed	0.0074	0.0085	0.0474 ***	0.0123	0.0343 *	0.0137	0.0164	0.0110
Rural resident	−0.0712 ***	0.0079	−0.0894 ***	0.0108	−0.0651 ***	0.0121	−0.0660 ***	0.0095
Age-fixed effects	Yes		Yes		Yes		Yes	
Region-fixed effects	Yes		Yes		Yes		Yes	
Survey-fixed effects	Yes		Yes		Yes		Yes	
Mean of the dependent variable	0.085		0.234		0.404		0.201	
Observations	14,278		14,285		14,281		17,479	

Notes: *** Significant at 1% level; * significant at 10% level.

**Table 5 ijerph-20-05048-t005:** The effect of SARS-CoV-2 pandemic on lost incomes in Ethiopia.

	Lost Household Income		Lost Business Income		Lost Farm Income		Lost Remittances	
Specification	β	SE	β	SE	β	SE	β	SE
Female-headed family	0.0273 **	0.0095	0.0129 ***	0.0031	−0.0035	0.0197	0.0141 ***	0.0032
Age-fixed effects	Yes		Yes		Yes		Yes	
Region-fixed effects	Yes		Yes		Yes		Yes	
Survey-fixed effects	Yes		Yes		Yes		Yes	
Mean of the dependent variable	0.388		0.024		0.353		0.019	
Observations	17,453		17,479		4279		17,479	

Notes: *** Significant at 1% level; ** significant at 5% level. All the regression models included controls for the level of education, marital status, and place of residence (urban vs. rural).

## Data Availability

The data used for this study are publicly available upon a formal request to the World Bank. Data for Kenya can be requested here: https://microdata.worldbank.org/index.php/catalog/3774/get-microdata (8 March 2023). Data for Ethiopia can be requested here: https://microdata.worldbank.org/index.php/catalog/3716/data-dictionary (8 March 2023).

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
