# Peer review of "Leaving No Women Behind: Evaluating the Impact of the COVID-19 Pandemic on Livelihood Outcomes in Kenya and Ethiopia"

_ijerph, 2023, doi:10.3390/ijerph20065048_

Round 1
Reviewer 1 Report
The manuscript, entitled “Leaving no women behind: Evaluating the impact of the COVID-19 pandemic on livelihood outcomes in Kenya and Ethiopia”, intends to investigate and evaluate the impact of the COVID-19 pandemic on livelihood outcomes inclusive of income, consumption, and food insecurity in Kenya and Ethiopia. The main observation made by the authors on the basis of the analysis is that the pandemic was associated the income losses and increased food insecurity, particularly in households with female-headed family most likely to increase in both in Kenya and Ethiopia.
However, in my opinion, there are several points for the authors that need to be addressed before acceptance:
· The introduction tells the prospective readers what they can expect, but in order to keep the manuscript consistent, I suggest authors to combine the Introduction section with the COVID-19 pandemic, gender, and outcomes, and also remove repetition of the intent of the study on lines 72-78, and lines 110-111.
· If representing numerical values by rounding off, then maintain a consistent flow throughout the entire manuscript. For instance, the numerical value on line 268 is represented by rounding off, although most of the values in the manuscript are not rounded off.
· Kindly mention the limitations of this study design and the acquired data sets.
Author Response
RESPONSES TO REVIEWER 1 COMMENTS
We are extremely grateful to all the reviewers for taking their valuable time reviewing our paper. They all highlighted and provided some useful suggestions which we believe significantly improved the quality of the paper. However, in few instances where we did not agree with the reviewers’ comments, we respectfully provided some reasons to that effect. Again, thank you so much.
REVIEWER 1
The manuscript, entitled “Leaving no women behind: Evaluating the impact of the COVID-19 pandemic on livelihood outcomes in Kenya and Ethiopia”, intends to investigate and evaluate the impact of the COVID-19 pandemic on livelihood outcomes inclusive of income, consumption, and food insecurity in Kenya and Ethiopia. The main observation made by the authors on the basis of the analysis is that the pandemic was associated the income losses and increased food insecurity, particularly in households with female-headed family most likely to increase in both in Kenya and Ethiopia.
However, in my opinion, there are several points for the authors that need to be addressed before acceptance:
- The introduction tells the prospective readers what they can expect, but in order to keep the manuscript consistent, I suggest authors to combine the Introduction section with theCOVID-19 pandemic, gender, and outcomes, and also remove repetition of the intent of the study on lines 72-78, and lines 110-111.
- We thank the reviewer for their useful comments and suggestions.
- We do agree that lines 110-111 are somewhat duplicated. We have removed the duplicated content in the revised version of the paper.
- However, we disagree with the reviewer’s suggestion here to combine the introduction and section 2 (which gives a brief summary of the transmission mechanism or how the pandemic is likely to influence food insecurity outcomes). By combining these sections, we think that will disturb the flow of the paper as we had not written these sections with the goal of having it as one section. Thus, to the reader, the introduction might unnecessarily be too lengthy and not very useful to them.
- If representing numerical values by rounding off, then maintain a consistent flow throughout the entire manuscript. For instance, the numerical value on line 268 is represented by rounding off, although most of the values in the manuscript are not rounded off.
- Thank you for your comments. We have now carefully checked this throughout the manuscript.
- Kindly mention the limitations of this study design and the acquired data sets.
- We agree with the reviewer and have now included a paragraph under the discussion section on the limitations of the analysis.

Reviewer 2 Report
Abstract
· Please provide further information about the study outcomes
· In the abstract you need to answer the following questions, what, why and how and discuss the study new findings, limitations, and future research
· The abstract should state briefly the purpose of the research, the principal results and major conclusions. An abstract is often presented separately from the article, so it must be able to stand alone
Introduction
- discuss the research aims, research gap and discuss the paper layout Add up-to-date references to support your discussion
- The necessity and innovation of the article should be presented to the introduction
The literature reviewed and cited is in the main rather old
Methods
· The methodology of this study should be detailed, limit information was provided on method and materials.
· Sampling strategy should be defined, how the samples were collected
· The authors should be able to control/reduce the selection bias
· Sample size calculation is missing
· Validation of the study questionnaire is missing in this study which affect the internal and external validity.
Discussion
- I believe that more in depth discussion is needed. The discussion as present now is simple and concise. Revision of more papers using similar technique is needed
- In the discussion, please discuss if the study research questions are answered or not Also introduce the model in detail. Draw a conclusion from this study and present the limitations and future research.
- . The major defect of this study is the debate or Argument is not clear stated in the introduction session. Hence, the contribution is weak in this manuscript. I would suggest the author to enhance your theoretical discussion and arrives your debate or argument
- Please make sure your conclusions' section underscore the scientific value added of your paper, and/or the applicability of your findings/results, as indicated previously.
- Please revise your conclusion part into more details. Basically, you should enhance your contributions, limitations, underscore the scientific value added of your paper, and/or the applicability of your findings/results and future study in this session.
Author Response
RESPONSES TO REVIEWER 2 COMMENTS
We are extremely grateful to the reviewer for taking their time reviewing this paper. The reviewer has highlighted and provided some useful suggestions, some of which we believe significantly improved the quality of our paper. However, in very few instances where we did not agree with the reviewers’ comments, we respectfully provided some reasons to that effect. Again, thank you for your service.
Abstract
- Please provide further information about the study outcomes
- Thank you for your comments.
- We actually thought that the outcome variables section (section 3.2) is pretty detailed since we have included the actual survey question verbatim.
- We have also clearly indicated how we have created the outcome variables.
- It would be good if the reviewer could specify what kind of additional information they are looking for.
- In the abstract you need to answer the following questions, what, why and how and discuss the study new findings, limitations, and future research
- Thank you for your comments / suggestions.
- In light of the comments from reviewer 3, we don’t think that we need to change anything in the abstract, apart from minor grammatical corrections.
- We disagree with the reviewer here since answering all that information will be too much for the abstract.
- Also, we disagree with the reviewer because we are unclear exactly what the reviewer means when they say “need to answer the following questions, what, why, and how and discuss the study new findings, and limitations, and future research”. The reviewer has not stated any question/(s) to be answered in their comments. We don’t think it will be a good idea also to showcase the limitations of the findings in the abstract. The limitations of the study have been included in the discussion section of the paper.
- In the abstract, we think that by providing a brief background, stating the goal of our research, what data we use, what we found, and implications for policy, is enough for an abstract.
- The abstract should state briefly the purpose of the research, the principal results and major conclusions. An abstract is often presented separately from the article, so it must be able to stand alone
- We certainly agree with the reviewer’s insights and comments here.
- This is what we have included in our abstract.
Introduction
- Discuss the research aims, research gap and discuss the paper layout Add up-to-date references to support your discussion
- We thank the reviewer for their time reviewing our paper.
- However, we are unclear as to what the reviewer wants us to change or add in the introduction section. The reviewer’s comment is rather general and very hard to know what we must do in this case. It would have been more useful if the reviewer had noted a few things that needs attention and perhaps give a few tips or suggestions as to how we might go about addressing those issues.
- All the literature we cited in the introduction [references 1-9] is from 2020-2023.
- The necessity and innovation of the article should be presented to the introduction
- The innovation of the article is clearly articulated in the introduction (see lines 41-44).
- We are unclear on what the reviewer means by “necessity”. If by “necessity” the reviewer meant relevance of the study, then this has clearly been articulated as well (see first paragraph of the introduction). Moreover, research on the impact of the pandemic is still growing and no one knows yet exactly how the pandemic is disproportionately impacting livelihoods.
- The literature reviewed and cited is in the main rather old
- No, we respectfully disagree with the reviewer here. The literature cited in the introduction was published between 2020 and 2023. We invite the reviewer to check our reference list. Also, we are unclear on what the reviewer means when they say “…cited is in the main rather old”?
Methods
- The methodology of this study should be detailed, limit information was provided on method and materials.
- Thank you for your review and comments.
- We are unclear here which aspect need to be more detailed under Materials and Methods?
- We think that we have provided lot of details about the data we are using, the measures (variables used and how we define these), the empirical strategy is also detailed. The empirical model is just a simple linear probability model and cannot be more detailed than what we have already done. We feel that the current model described is simple and clear to the reader and reflects what we have done in the paper and the results that we discuss.
- We are happy to provide more information if the reviewer can be more specific on which section requires additional information. Also, it would be useful as well if the reviewer can be more specific what sort of information, he/she feels was left out / is missing or needs to be included.
- Otherwise, we have not added anything new.
- Sampling strategy should be defined, how the samples were collected
- Thank you for your comments.
- In this paper, we are using secondary data. Hence, we have not collected any data for the purposes of the analysis.
- We have provided a citation to refer the reader to additional information as to how the World Bank and its partners have collected the data that we use in the paper.
- Detailed methodology and data collection techniques are described here: https://microdata.worldbank.org/index.php/catalog/3716/data-dictionary. We have included the links to the data already in the paper.
- The authors should be able to control/reduce the selection bias
- In the paper, we did not say there was selection bias. Instead, we noted that selection bias could “potentially” be an issue given that the data is only representative of households owning a working phone.
- We don’t know whether selection bias is actually a problem or not in this case since we have not formally tested it. We just flagged that as a potential caveat to the analysis. The source of the potential selection bias is clearly something we cannot do anything about. We have included this as a potential limitation to the analysis.
- Sample size calculation is missing
- We do agree with the reviewer here that a sample size calculation is missing from our study.
- In our case, we don’t think a sample size calculation was required. Here is why:
- Our sample is predetermined, as we use secondary level data. Also, we are not conducting some analysis on say a particular cohort or subgroup of the population where we need to determine the minimum required sample size. Instead, our analysis has a national focus and using the larger sample size available.
- We are not trying to detect a given effect size of some sort in this study. Typically, one does a sample size calculation to be able to detect a given effect size (at the specified levels of alpha and beta). That is not what we are doing in this research.
- Instead, we followed some guidelines / recommendations / rule-of-thumb suggested and applicable for quantitative empirical studies on the minimum recommended sample sizes for quantitative studies (for example, [1, 2]. The minimum recommended sample size to be able to detect precise and reliable estimates in quantitative studies should be greater than 200 and the larger the sample, the better.
- Validation of the study questionnaire is missing in this study which affect the internal and external validity.
- We thank the reviewer for their comment.
- Indeed, the reviewer is correct in saying that the “validation of the study questionnaire is missing” from this study and here is why that is the case:
- Validation of the questionnaire is NOT applicable in this study / in this case.
- We are using secondary data (data that has already been collected). The data custodians (in this case, the World Bank and its Partners) are responsible for validating their survey questionnaire used to collect their data – which I am sure they have already done.
- No action was done here in light of the reviewer’s comment.
Discussion
- I believe that more in depth discussion is needed. The discussion as present now is simple and concise. Revision of more papers using similar technique is needed
- Thank you for your observations.
- We have now re-worked the discussion section to reflect the reviewer’s suggestions.
- In the discussion, please discuss if the study research questions are answered or not Also introduce the model in detail. Draw a conclusion from this study and present the limitations and future research.
- Thank you for your review and comments.
- This has already been addressed in the discussion section. Right at the beginning of the discussion section, we started off by reminding the reader what the primary goal of the analysis was. We then briefly stated what we found, which essentially answers the reviewer’s concern regarding whether we answer the research questions or not.
- We have included a section regarding the limitations of this study.
- Also, we have provided some suggestions for future research in Ethiopia and Kenya.
- We disagree with the reviewer here, the discussion section is not the place to be “Introducing the model in detail” as suggested by the reviewer. The model has already been introduced in detail (see empirical strategy).
- The major defect of this study is the debate or Argument is not clear stated in the introduction session.
- Thank you for your comments.
- We disagree with the reviewer here. The primary goal of this study has been clearly stated.
- Hence, the contribution is weak in this manuscript. I would suggest the author to enhance your theoretical discussion and arrives your debate or argument
- Thank you for your review and comments. We have refined our arguments and make this clearer in the paper.
- We have clearly stated the primary goal of the paper and what the analysis is about. We have also included details on what previous studies have done, which also highlights the gap in the literature.
- Please make sure your conclusions' section underscore the scientific value added of your paper, and/or the applicability of your findings/results, as indicated previously.\
- Thank you for your review and comments. This is now addressed in the paper.
- Please revise your conclusion part into more details. Basically, you should enhance your contributions, limitations, underscore the scientific value added of your paper, and/or the applicability of your findings/results and future study in this session.
- Thank you for your review and comments. However, it’s unclear what the reviewer is asking that we do here.
- Typically, a conclusion section should be brief or concise and straight to the point. This is what we have done.
References used.
- Boo, S. and E.S. Froelicher, Secondary analysis of national survey datasets. Japan journal of nursing science, 2013. 10(1): p. 130-135.
- Perecman, E. and S.R. Curran, A handbook for social science field research: essays & bibliographic sources on research design and methods. 2006: Sage Publications.

Reviewer 3 Report
The authors have evaluated the impact of the COVID-19 pandemic on livelihood outcomes in Kenya and Ethiopia in their study. I would like to congratulate the authors for their work and for their analyze of the data provided from the surveys.
Abstract:
· The abstract is well written. Please write it at the past tense, as it summarizes your main findings from your study done into the past.
· The keywords are used accordingly to the subject of the paper.
Introduction:
· Line 32: I would suggest mentioning the strategy also as a bibliography entry.
· Line 36: coronavirus pandemic – coronaviruses are of many types. I suggest replacing it with the SARS-COV-2 pandemic (which causes the COVID-19 disease)
· Line 74: Please use the same term for the pandemic ( SARS-COV-2 pandemic)
· Line 74-78 – You do not have to present findings in this part – as you set the aim (object) of your study in this part.
· Also, I have noticed that the time use is the present one. As you said in line 75, the study was based on phone surveys, so it was done in the past and now you present the results. My suggestion is to keep the past tense in the paper.
· Line 87 – Please use the definition provided by WHO as a bibliography entry, at least the webpage. Also, any abbreviation should be defined in brackets at the first use in text.
· Line 110-111. COVID-19 still exists. Please rewrite the sentence – I think that you refer to the fact that you have analyzed during the COVID-19 pandemic.
Materials and methods:
· Line 116 – rapid response telephone survey - please set it as bibliography entry at leas from the World bank site or something else.
· Line 117 – are the surveys still working? Please specify.
· In the data part – I would suggest specifying better the period from which the data from this study is presented, the inclusion and exclusion criteria.
· As it is specified in line 116 the name for the survey in Kenya, please specify it also for Ethiopia.
· I have a commendation regarding the description of the measure chapter. It is well written, well explained and with specific data. I would suggest, as you refer to the questions used into the surveys for line 154-156, to make a supplementary file regarding the questions.
· I would recommend mentioning also the limitations of your study.
Results:
· As a general comment, in the results part you only show the results, but you don not interpretate them (for example to make a comparison between the age in Kenya and Ethiopia – line 268). The comparation and the discussion regarding your results is made into the discussion part.
· For table one – I suggest for the Kenya part – where you do not have any answers, you can write n/a in order to emphasize that is not assessed.
· Line 280 – 313 – You have very well presented the scientific idea and the hypothesis, but again, you are discussing the results and make conclusions based on your results in the RESULTS part – this is usually done in the discussion part.
· The explanation of the results is well done, rigorous and well presented. But you have to cut the part where you discuss your results in the results part.
Discussion:
· The discussion part is too short for this rigorous study. As the study is very well done, I suggest transferring the interpretation of your results from the results part to the discussion part, and to make a direct comparison between your results and the literature results.
Conclusion
· The conclusion summarizes the results and their interpretation.
Author Response
RESPONSES TO REVIEWER 3 COMMENTS
We are extremely grateful to the reviewer for taking their time reviewing this paper. The reviewer has highlighted very useful suggestions which has greatly enhanced the quality of our paper. Again, thank you for your service.
The authors have evaluated the impact of the COVID-19 pandemic on livelihood outcomes in Kenya and Ethiopia in their study. I would like to congratulate the authors for their work and for their analyze of the data provided from the surveys.
- We thank the reviewer for their positive comments and words of encouragement.
Abstract:
- The abstract is well written. Please write it at the past tense, as it summarizes your main findings from your study done into the past.
- We thank the reviewer for their positive comments.
- We have incorporated the reviewer’s suggestions and revised the abstract and the rest of the paper as needed.
- The keywords are used accordingly to the subject of the paper.
- We thank the reviewer for their positive comments and words of encouragement.
Introduction:
- Line 32: I would suggest mentioning the strategy also as a bibliography entry.
- Thank you for the comment. But, the strategy is already included as a bibliography entry. Please see line 32 and also the reference list. Reference #1 is the strategy.
- Line 36: coronavirus pandemic – coronaviruses are of many types. I suggest replacing it with the SARS-COV-2 pandemic (which causes the COVID-19 disease)
- We agree with the reviewer. We have now changed the text to reflect the reviewer’s suggestion.
- Line 74: Please use the same term for the pandemic ( SARS-COV-2 pandemic)
- Thank you. We have now changed the text to reflect the reviewer’s suggestion.
- Line 74-78 – You do not have to present findings in this part – as you set the aim (object) of your study in this part.
- Thank you. We have now changed the text to reflect the reviewer’s suggestion.
- Also, I have noticed that the time use is the present one. As you said in line 75, the study was based on phone surveys, so it was done in the past and now you present the results. My suggestion is to keep the past tense in the paper.
- Thank you. We have now changed the text to reflect the reviewer’s suggestion.
- Line 87 – Please use the definition provided by WHO as a bibliography entry, at least the webpage. Also, any abbreviation should be defined in brackets at the first use in text.
- Thank you, we have now included this definition as a bibliography entry and have also defined WHO at first usage.
- Line 110-111. COVID-19 still exists. Please rewrite the sentence – I think that you refer to the fact that you have analyzed during the COVID-19 pandemic.
- We do agree with the reviewer. We have now removed the sentence completely as it was also duplicated.
Materials and methods:
- Line 116 – rapid response telephone survey - please set it as bibliography entry at leas from the World bank site or something else.
- Thank you, we have now provided a bibliography entry to this. We have cited the World Bank website where we downloaded the dataset from.
- Line 117 – are the surveys still working? Please specify.
- From our understanding, the datasets have been collected already.
- For Kenya, the last wave available, i.e., wave 8 was completed on 8th of July 2022. We have reflected this information in the paper.
- In the data part – I would suggest specifying better the period from which the data from this study is presented, the inclusion and exclusion criteria.
- We have provided additional information to reflect what data we use.
- In summary, for both countries, we have pre-COVID and Post-COVID data.
- There are 8 post-COVID waves/rounds of data for Kenya, of which we use 6 waves out of the total. Thus, the 6 waves comprise of the period between 14 May 2020 to 3 November 2021.
- For Ethiopia, we have 12 rounds of post-COVID data available of which we have used the first 6 waves of data. Thus, the 6 waves of data comprise of the period starting 22 April 2020 (round 1) to 14 October 2020 (round 6).
- The decision to use the first 6 rounds of data is primarily driven by the availability of outcome variables used for the analysis. As you be aware, the high frequency surveys were not consistently collecting the same information across countries. This is one of the limitation of the dataset.
- As it is specified in line 116 the name for the survey in Kenya, please specify it also for Ethiopia.
- This has been done as well. Thank you for noting this.
- I have a commendation regarding the description of the measure chapter. It is well written, well explained and with specific data. I would suggest, as you refer to the questions used into the surveys for line 154-156, to make a supplementary file regarding the questions.
- We thank the reviewer for their positive feedback. We have included an appendix for the questions used in the analysis.
- Thank you again for your insights.
- I would recommend mentioning also the limitations of your study.
- Again, we thank the reviewer for their insights and valuable suggestions.
- We have included a paragraph with limitations to the study.
Results:
- As a general comment, in the results part you only show the results, but you don not interpretate them (for example to make a comparison between the age in Kenya and Ethiopia – line 268). The comparation and the discussion regarding your results is made into the discussion part.
- Thank you for your observations. We have re-worked the results section to reflect the reviewer’s suggestion.
- For table one – I suggest for the Kenya part – where you do not have any answers, you can write n/a in order to emphasize that is not assessed.
- Thank you, this is done.
- Line 280 – 313 – You have very well presented the scientific idea and the hypothesis, but again, you are discussing the results and make conclusions based on your results in the RESULTS part – this is usually done in the discussion part.
- Thank you for your observations. We have re-worked the results section to reflect the reviewer’s suggestion.
- The explanation of the results is well done, rigorous and well presented. But you have to cut the part where you discuss your results in the results part.
- Thank you for your observations. We have re-worked the results section to reflect the reviewer’s suggestion.
Discussion:
- The discussion part is too short for this rigorous study. As the study is very well done, I suggest transferring the interpretation of your results from the results part to the discussion part, and to make a direct comparison between your results and the literature results.
- We are very grateful to the reviewer for their overall positive comments and support.
- We do agree with the reviewer’s suggestions and have edited the paper to reflect the reviewer’s input.
Conclusion
- The conclusion summarizes the results and their interpretation.
- Thank you for your positive comments and reviewer. We are very grateful for your support.

Round 2
Reviewer 2 Report
Dear Authors, thank you for addressing the reviewer comments, still critical points not addressed in the method section as following:
the Authors used a telephonic survey for data collection, then how they define the sampling frame, sample size?
How the Authors control for selection and response bias
How they evaluated the internal and external validity ?
Author Response
RESPONSES TO REVIEWER 2 COMMENTS
Dear Authors, thank you for addressing the reviewer comments, still critical points not addressed in the method section as following:
- We thank the reviewer again for their comments on our paper.
The Authors used a telephonic survey for data collection, then how they define the sampling frame, sample size?
- The reviewer is incorrect when they say that “The authors used a telephonic survey for data collection”
- The authors did not collect this data, as the reviewer seem to suggest.
- Instead, we have used already existing data or secondary data which we have described and provided complete citations to in the main paper.
- The data section of the paper provides details on how the World Bank has collected this data.
- We have also included these links in the paper to refer the reader to detailed methodology used in collection of the data. Found here for Kenya: https://microdata.worldbank.org/index.php/catalog/3774/get-microdata. And here for Ethiopia: https://microdata.worldbank.org/index.php/catalog/3716/data-dictionary
How the Authors control for selection and response bias
- We thank the reviewer for this comment.
- We agree with the reviewer that selection and response bias is a huge problem in telephone surveys in general. Indeed the World Bank has conducted some studies to test this. In a recent study, Ambel, McGee [1] tested this and found that to be an issue in the high-frequency phone surveys conducted in Ethiopia and other countries (see their paper in the references section). One of their recommendations to control for such biases was to use sampling weights.
- In our study, perhaps we had not clarified this in our description. We have incorporated sampling weights to minimise these biases, as Ambel, McGee [1] However, while Ambel, McGee [1] suggest that weight adjustment reduces bias in the interviewed samples, they also acknowledge that it is impossible to eradicate all the biases.
- We have revised the paper and made this explicitly clear that we applied the sampling weights to control for response bias as detailed in Ambel, McGee [1] (see the methods section on page 4, lines 166-169.
- Thanks again for your insights.
How they evaluated the internal and external validity?
- The aspects of internal and external validity certainly comprise of various aspects of the research process including, data collection methodology, study designs, ethical considerations, context of the research, interpretation of findings, among several other aspects.
- We conducted this study within the “good-practice” guidelines and recommendations / considerations documented in Fell, Pagel [2] for research on the impacts of COVID-19 in energy and social science research.
- From the previous review (Round 1), the reviewer has said: “Validation of the study questionnaire is missing in this study which affect the internal and external validity”
- If the reviewer still insists that we validate the questionnaire as per their R1 comments, the authors maintain that as we did not collect this data, we are not in a position to validate the questionnaire/(s) used by the World Bank to collect their data.
- Our research has been conducted following recommended guidelines for conducting social science research. Our results are also consistent with several other related research that has been conducted so far in different countries across the world regarding the impact of COVID-19 on food insecurity.
- The authors have also noted this as a potential limitation to the study (see limitations section).
- Specifically, we have included the following text in lines 555-562 to incorporate the reviewer’s concerns:
Lastly, as the SARS-COV-2 pandemic has challenged every sphere of our lives, including how we conduct empirical research, we acknowledge that questions on internal and external validity will always be necessary. As indicated in previous research, “The magnitude and speed, and reach of the changes to our lives are of a different order to anything that most people today have experienced” [2]. The implications of the SARS-COV-2 pandemic on the representativeness of research findings should be considered. In this research, we have followed the `best-practice’ recommended principles and guidelines to minimize the threats to the internal and external validity of the conclusions generated from this research.”
References used
- Ambel, A., K. McGee, and A. Tsegay, Reducing bias in phone survey samples. 2021.
- Fell, M.J., et al., Validity of energy social research during and after COVID-19: challenges, considerations, and responses. Energy Research & Social Science, 2020. 68: p. 101646.

Reviewer 3 Report
Dear authors,
All my suggestions were answered. Well done and congrats!!
No further comments to add.
Author Response
The authors thank the reviewer for their invaluable support and comments that improved the quality of the paper. Thank you for being so supportive.